

# Spatiotemporal evaluation of EMEP4UK-WRF v4.3 atmospheric chemistry transport simulations of health-related metrics for NO₂, O₃, PM₁₀ and PM₂.₅ for 2001-2010

C. Lin[1], M .R. Heal[1], M. Vieno[2], I. A. MacKenzie[3], B. G. Armstrong[4], B.K. Butland[5], A. Milojevic[4], Z. Chalabi[4], R. W. Atkinson[5], D. S. Stevenson[3], R. M. Doherty[3], P. Wilkinson[4]

[1]School of Chemistry, University of Edinburgh, Edinburgh, UK
[2]NERC Centre for Ecology & Hydrology, Penicuik, UK
[3]School of GeoSciences, University of Edinburgh, Edinburgh, UK
[4]Department of Social and Environmental Health Research, London School of Hygiene and Tropical Medicine, London, UK
[5]Population Heath Research Institute and MRC-PHE Centre for Environment and Health, St George's, University of London, London, UK

*Correspondence to*: M. Heal (m.heal@ed.ac.uk)

**Abstract**

This study was motivated by the use in air pollution epidemiology and health burden assessment of data simulated at 5 km × 5 km horizontal resolution by the EMEP4UK-WRF v4.3 atmospheric chemistry transport model. Thus the focus of the model-measurement comparison statistics presented here was on the health-relevant metrics of annual and daily means of NO₂, O₃, PM₂.₅ and PM₁₀ (daily maximum 8-hour running mean for O₃). The comparison was temporally and spatially comprehensive covering a 10-year period (2 years for PM₂.₅) and all measurement data from the UK national reference monitor network, which applies consistent operational and QC/QA procedures for each pollutant (60, 49, 29 and 35 sites for NO₂, O₃, PM₂.₅ and PM₁₀, respectively). The two most important statistics highlighted in the literature for evaluation of air quality model output against policy (and hence health)-relevant standards – correlation and bias – were evaluated by site type, year, month and day-of-week. Model-measurement correlation and bias were generally better than values found in past studies that allowed for measurement uncertainties. Temporal correlations of daily concentrations were good for O₃, NO₂ and PM₂.₅ at both rural and urban background sites (median values of $r$ across sites in the range 0.70-0.76 for O₃ and NO₂, and 0.65-0.69 for PM₂.₅), but poorer for PM₁₀ (0.47-0.50). Bias differed between environments, with generally less bias at the background sites and least bias at rural background sites (median normalised mean bias (NMB) values for daily O₃ and NO₂ of 8% and 11%, respectively). At urban background sites there was a negative model bias for NO₂ (median NMB = −29%) and PM₂.₅ (−26%) and a positive model bias for O₃ (26%). The directions of these biases are consistent with expectations of the effects of averaging primary emissions across the 5 km × 5 km model grid in urban areas, compared with monitor locations that are more influenced by these emissions than the grid average. This effect was particularly pronounced for comparison against urban traffic monitors, which are deliberately located close to strong sources of NOₓ and PM. The biases are also indicative of potential underestimations of primary NOₓ and PM emissions in the model, and, for PM, with known omissions in the model of some PM components, e.g. wind-blown dust. There were instances of monthly and weekday/weekend variations in extent of model-measurement bias. Overall, the greater uniformity in temporal correlation than in bias is strongly indicative that the main driver of model-measurement differences (aside from grid vs monitor spatial representativity) was inaccuracy of model emissions (both in annual totals and in the monthly and day-of-week temporal factors applied in the model to the totals) rather than simulation of atmospheric chemistry and transport processes. Since, in general for epidemiology, capturing correlation is more important than bias, the detailed analyses presented here support the use of data from this model framework in air pollution epidemiology.





## 1 Introduction

The adverse associations between ambient air pollution – especially particulate matter (PM), ozone ($O_3$) and nitrogen dioxide ($NO_2$) – and morbidity and mortality are well documented (WHO, 2006; WHO, 2013b; WHO, 2013a). Air pollution also

causes substantial environmental and economic impact to ecosystems and crops (ROTAP, 2009; LRTAP Convention, 2010; Harmens et al., 2015).

Whilst policies and legislation have been put in place to limit and mitigate the impacts of air pollution (Heal et al., 2012), there is increasing recognition that more effective protection of human health may be achieved by not focusing on individual

pollutants but by taking a multi-pollutant approach (USEPA, 2008; Dominici et al., 2010). Compared with the traditional single pollutant focus (WHO, 2006), an approach based on pollution mixtures has the advantage of enabling the complexity of exposures and health effects to be characterized more fully: it can help identify harmful emission sources, and it has potential to provide a more effective framework for air-quality regulation, for example by focusing on sources and pathways that influence several pollutants at once. There are analytical complexities in assessing the potential interactions between

combinations of pollutants (Kim et al., 2007; Mauderly and Samet, 2009), including the paucity of measured exposure data, which are typically derived from relatively sparse monitoring sites that may measure different combinations of pollutants at different locations. Furthermore, monitor networks are usually established for compliance with legislation (e.g. deliberately sited close to, and away from, pollution sources), so may lack representativeness for characterising population exposure (Duyzer et al., 2015) leading to bias in air pollution epidemiology (Sheppard et al., 2012).

Modelling can increase the availability of air pollution data (Jerrett et al., 2005). The current gold standard for air-quality modelling are process-based, deterministic atmospheric chemistry models (Colette et al., 2014). These seek to simulate the multitude of complex factors that govern the spatial and temporal variability in air pollutant concentrations, including the distributions of different emissions sources, local and long-range dispersion processes, in situ photochemistry and dry and wet

deposition processes.

As part of a multi-institution project, we have undertaken epidemiological studies on the health impacts of exposure to multiple pollutants using UK-wide distributions of surface air pollution at hourly temporal resolution over multiple years (2001-2010), at 5 km × 5 km horizontal resolution, derived from the EMEP4UK-WRF atmospheric chemistry transport model (ACTM).

This represents a unique dataset of ACTM simulations at this spatial and temporal resolution over this geographical coverage and time duration. The EMEP4UK-WRF model (Vieno et al., 2010; 2014; 2016) is a regional application of the European Monitoring and Evaluation Programme (EMEP) MSC-W model (Simpson et al., 2012). The EMEP model framework has been evaluated and used for many years in scientific support (Fagerli et al., 2015), in, for example, evaluation of emissions regulations within the UNECE framework (e.g. the Gothenburg Protocol) and the European Commission's Clean Air for

Europe (CAFE) programme (www.emep.int).

The high temporal and spatial resolution output from the EMEP4UK-WRF model has many advantages for air pollution studies including: (i) provision of data at times and locations where monitoring data are not available; this has the dual benefit of increasing effective sample size in multi-pollutant health epidemiology and of reducing reliance on the assumption that a single

monitor is representative of species concentrations over a large area; (ii) provision of data on individual particle chemical components in addition to the aggregated mass concentration of PM that is measured; (iii) the facility to explore many related





aspects such as geographical or demographic differences in exposures to air pollutant mixtures (and related issues of environmental justice), and (iv) the impacts of potential future emissions scenarios.

It is important to have an understanding of the performance capabilities of any model, relevant to the use to which the model
output is to be put. Much has been written on air quality model evaluation (see, for example, Vautard et al., 2007; Dennis et al., 2010; Derwent et al., 2010; Rao et al., 2011; Thunis et al., 2012; Thunis et al., 2013; Pernigotti et al., 2013), including publications arising out of international collaborative programmes such as AQMEII (Air quality modelling evaluation international initiative, http://aqmeii-eu.wikidot.com) and FAIRMODE (Forum for air quality modelling in Europe, http://fairmode.jrc.ec.europa.eu). The literature ranges from discussion of epistemological categories of evaluation to
development of specific metrics and criteria for comparison between modelled and measured concentrations. Detail is not repeated here, other than to note that there are fundamental limitations to agreement between model and measurements, which include: uncertainties intrinsic to the measurements; limitations in model input data (e.g. emissions) and in other aspects of model descriptions of physical processes; and that models simulate a volume-average concentration whilst monitors measure at a specific location.

The objective of this paper is to record detailed assessment of the modelled surface concentrations of $O_3$, $NO_2$ and $PM_{2.5}$ and $PM_{10}$ using metrics of these pollutants relevant to air pollution epidemiology and health burden assessment, namely the daily mean for PM and $NO_2$ and the maximum daily 8-h running mean for $O_3$. The measurements are taken from the UK's Automatic Urban and Rural Network (AURN) of 'real-time' reference monitors. The key emphasis in this work is comprehensiveness
and consistency: the model-measurement evaluation is UK wide, over an extended time period (10 years), and based on measurements subject to a single set of operational and QC/QA procedures for each pollutant.

## 2. Methodology

### 2.1. Model data

The EMEP MSC-W regional Eulerian ACTM is described in Simpson et al. (2012) and at www.emep.int. The EMEP4UK model providing data in this work (Vieno et al., 2014; Vieno et al., 2016) was based on version vn4.3, driven by meteorology from the Weather Research and Forecast model (www.wrf-model.org) version 3.1.1. The WRF model was constrained by boundary conditions from the US National Center for Environmental Prediction (NCEP)/National Center for Atmospheric Research (NCAR) Global Forecast System (GFS) at 1° resolution, every 6 hours. Nesting within the EMEP4UK model reduces
horizontal resolution from 50 km × 50 km over a greater European model domain to 5 km × 5 km over an inner domain covering the British Isles plus adjacent parts of France, Belgium, Holland and Denmark, as illustrated in Vieno et al. (2014). Both WRF and EMEP4UK models use 20 vertical layers, with terrain following coordinates, and resolution increasing towards the surface (centre of the surface layer ~45 m). The vertical column extends up to 100 hPa (~16 km). The boundary conditions for the inner domain were taken from 3-hourly output from the European domain in a one-way nested setup, whilst for the
European domain they were measurement derived and adjusted monthly (Vieno et al., 2010). Ground-level modelled species concentrations were calculated hourly at 3 m above the surface vegetation or other canopy by making use of the constant-flux assumption and definition of aerodynamic resistance (Simpson et al., 2012).

Anthropogenic emissions of $NO_x$, $NH_3$, $SO_2$, primary $PM_{2.5}$, primary $PM_{coarse}$ (where $PM_{coarse}$ is the difference between $PM_{10}$
and $PM_{2.5}$), CO and non-methane VOC for the UK for each modelled year were taken from the National Atmospheric Emission Inventory (NAEI, http://naei.defra.gov.uk) at 1 km² resolution and aggregated to 5 km × 5 km resolution. For the outer domain,





the model used the EMEP 50 km × 50 km resolution emission estimates provided by the Centre for Emission Inventories and Projections (CEIP, http://www.ceip.at/). The annual total emissions were temporally split using prescribed monthly, day-of-week, and diurnal hourly emission factors (the latter differing between weekdays, Saturday and Sundays) for each pollutant and for each of the SNAP (Selected Nomenclature for Sources of Air Pollution) sectors (Simpson et al., 2012). Methane

5  concentration was prescribed. Emissions estimates for international shipping were those from ENTEC UK Ltd. (now Amec Foster Wheeler) (ENTEC, 2010). Daily emissions from biomass burning were derived from the Fire INventory from NCAR version 1.0 (FINNv1) (Wiedinmyer et al., 2011). Natural emissions of isoprene, monoterpenes, dimethylsulfide (DMS), wind-induced sea salt and $NO_x$ from soils and lightning, were as described in Simpson et al. (2012). Natural emissions of dust included Saharan dust uplift, but not of windblown dust within the model domain.

The default EMEP MSC-W photochemical scheme was used, which contains 72 gas-phase species and 137 reactions; the gas/aerosol partitioning formulation was the model for aerosols reacting system (MARS) (Binkowski and Shankar, 1995). Simulation of secondary organic aerosol (SOA) formation, ageing and partitioning was via the 1-D volatility basis set (Donahue et al., 2006) with its implementation in the model as described by Bergström et al. (2012). The EMEP4UK model

output for $PM_{2.5}$ comprised the sum of the $PM_{2.5}$ fractions of: elemental carbon (EC), 'other' primary PM in the emissions inventories (encompasses material such as flyash, and brake and tyre wear), sea salt, mineral dust, primary and secondary organic matter (OM), ammonium ($NH_4^+$), sulphate ($SO_4^{2-}$) and nitrate ($NO_3^-$). $PM_{10}$ is the sum of $PM_{2.5}$ plus the $PM_{coarse}$ fractions of EC, 'other' primary PM (as above), sea salt, dust, OM and $NO_3^-$. The split of $NO_3^-$ into $PM_{coarse}$ and $PM_{2.5}$ uses a parameterised approach dependent on relative humidity, as described by Simpson et al. (2012). It is acknowledged this split is

somewhat uncertain, as discussed in Vieno et al. (2014). Despite the comprehensiveness of PM composition simulation, some known contributions are missing, in particular wind-blown dust. Also, as described in the next section, different measurement techniques and conditions incorporate different proportions of the ambient PM water content. Because of uncertainty in what measurements measure, and variability in measurement techniques employed through the time period of interest, we chose to use as model output the dry mass of PM. This contributes some unquantifiable variable negative model bias for $PM_{2.5}$ and

$PM_{10}$.

## 2.2. Measurement data

Hourly measurements of the concentrations of $NO_2$, $O_3$, $PM_{10}$ and $PM_{2.5}$ at the AURN stations during 2001-2010 were downloaded and processed using the R package 'openair' (Carslaw and Ropkins, 2012) from the R workspaces provided and

updated daily by Ricardo-AEA. Because of the emphasis in this study on data for health-related applications, the model-measurement comparisons were principally based on the daily pollutant metrics recommended by the World Health Organisation (WHO, 2006), i.e., daily mean concentrations for $NO_2$, $PM_{2.5}$ and $PM_{10}$ ($NO_2$_daymean, $PM_{2.5}$_daymean and $PM_{10}$_daymean), and daily maximum running 8-h mean for $O_3$ ($O_3$_max8hmean).

A data capture threshold of 75% was applied throughout the process of calculating statistics from the hourly measurements, as is standard protocol for EU data reporting (http://acm.eionet.europa.eu/databases/airbase/aggregation_statistics.html). For example, daily mean concentrations of $NO_2$, $PM_{2.5}$ and $PM_{10}$ were only calculated when there were at least eighteen hourly measurements in a day. For $O_3$, there had to be at least six hourly measurements in any 8-h window for an 8-h rolling mean to be calculated, and at least eighteen 8-h rolling means for a daily maximum 8-h mean to be valid.

Comparison with model output was only undertaken for AURN sites with ≥75% data capture rate over the whole 10-y period. This means that at least 2,739 out of 3,652 pairs of daily measured and modelled values were required for inclusion. For $PM_{2.5}$,



there were only four sites meeting the 75% data capture requirement over the ten years, so comparisons for $PM_{2.5}$ were restricted to the period 2009-2010.

AURN monitoring sites are classified according to their general location and proximity to particular sources of air pollution (https://uk-air.defra.gov.uk/networks/site-types). Sites classified as suburban background (only one or two sites per pollutant), suburban industrial (one site) and urban industrial (four sites or fewer depending on pollutant) were excluded from the model-measurements comparison as being insufficient in number to provide meaningful comparison for these site classifications. Model-measurement comparison therefore focused on potential differences between rural background (RB), urban background (UB) and urban traffic (UT) sites. The numbers of each type of AURN site contributing data to this model-measurement comparison are summarised in Table 1. The names, coordinates, classifications and pollutant data captures of all sites supplying data for this work are given in Supplementary Information Table S1.

The coordinates of each AURN station with valid measurements during the period 2001-10 was used to locate the 5 km × 5 km grid of the EMEP4UK domain whose centroid was closest to the station. The WRF-modelled hourly 2-m surface temperature data at each AURN site were also extracted and converted to daily means.

Measurements from the UK AURN adhere to EU Directives on reference instrumentation and QA/QC procedures. Concentrations of $NO_2$ and $O_3$ are derived from chemiluminescence and UV-absorption analysers, respectively. The 'real time' measurement of PM mass concentrations is technically more challenging than for $O_3$ and $NO_2$, and the instrumentation used in the UK varied during the 2001-10 period. After about 2008, the majority of measurements of $PM_{10}$ and $PM_{2.5}$ have been made by TEOM-FDMS (Tapered Element Oscillating Microbalance Filter Dynamics Measurement System) which has been demonstrated as equivalent to the EU reference method (Harrison, 2010). The TEOM-FDMS system records a value for both 'volatile' and 'non-volatile' PM and it is the sum of these values that is used in this work. All the 2009-10 $PM_{2.5}$ measurement data in this study are derived from TEOM-FDMS instruments. However, for $PM_{10}$, prior to the introduction of the auxiliary FDMS unit, measurements were derived using the TEOM instrument alone. The inlet and element of these instruments were held at 50 °C to limit condensation of water, but this caused loss of some volatile components of $PM_{10}$. All TEOM values were therefore multiplied by 1.3 before archiving to provide an estimate of the average loss of volatile components, as recommended by the EC Working Group on Particulate Matter (EC, 2001). $PM_{10}$ values from the few TEOM-only instruments remaining in the AURN after the general introduction of FDMS units in 2008 have been scaled using the more sophisticated Volatile Correction Model (Green et al., 2009), rather than the single 1.3 scaling factor, to account for the loss of volatile components. $PM_{10}$ data from the few Beta-Attenuation Monitor (BAM) instruments present in the AURN have been scaled by 1.3 if they had a heated inlet and 0.83 if they did not have a heated inlet.

The objective of all these external scaling processes for these PM measurements has been to provide the best practical measure of 'reference equivalent' $PM_{10}$ (and $PM_{2.5}$) mass concentrations spatially and temporally across the AURN. Nevertheless, these instrumental issues introduce considerable additional uncertainty to the PM measurement data: first, scaling factors, where applied, are an average scaling in time and space whereas the real scaling that would have been required would have varied between sites and for different times at an individual site; secondly, there may be a discontinuity in the $PM_{10}$ time series associated with instrument change at a particular site, and dates of instrument change varied across the network. Uncertainty in measurement-model comparison is also introduced by the use of dry mass PM as the model output.

Irrespective of these changes to $PM_{10}$ instrumentation, all PM, $NO_2$ and $O_3$ instruments in the AURN are maintained and calibrated in accordance with the QA/QC protocol for the UK ambient air quality monitoring network (http://uk-





air.defra.gov.uk/networks/network-info?view=aurn), and all data are subject to the network data review and ratification process before 'ratified' archiving.

### 2.3. Evaluation of spatial aspects of model performance

The coherence between long-term spatial patterns of modelled and measured concentrations was investigated through the correlation across sites of the 10-y (2-y for $PM_{2.5}$) means of the daily pollutant metrics at each site.

### 2.4. Evaluation of temporal aspects of model performance

The daily pollutant metrics were grouped by day of week, month of year, and year of the 10-y period. Statistics were then
calculated on the grouped pairs of daily model simulations and measurements for each pollutant at each site, and summarised by site type.

Of the various statistics proposed for quantifying performance of air-quality models, correlation and bias are consistently cited for evaluation against policy-relevant metrics of pollutant concentration (USEPA, 2007; Derwent et al., 2010; Thunis et al.,
2012). Thus, in this study the following statistics were calculated. (In each of the following, the index $i$ runs over the $n$ pairs of model ($M_i$) and observation ($O_i$) concentrations per time series at each site. The term 'observation' is used, in this section only, synonymously with the term 'measurement' used elsewhere in this paper, to avoid ambiguity of an $M$ label for model and for measurement.)

- Pearson's correlation coefficient ($r$):

$$r = \frac{1}{n-1} \sum_{i=1}^{n} \left( \frac{M_i - \bar{M}}{s_M} \right) \left( \frac{O_i - \bar{O}}{s_O} \right)$$

$\bar{M}$ and $\bar{O}$ are the mean of the modelled and observed concentrations respectively, and $s_M$ and $s_O$ are their respective sample standard deviations.

- Mean bias (MB) and normalised mean bias (NMB):

$$MB = \frac{1}{n} \sum_{i=1}^{n} M_i - O_i \qquad \text{and} \qquad NMB = \frac{\sum_{i=1}^{n} M_i - O_i}{\sum_{i=1}^{n} O_i}$$

- FAC2, the proportion of all pairs of modelled and observed concentrations that are within a factor of two of each other. This statistics provides additional general indication of overall model skill.

### 3. Results

### 3.1. Evaluation of spatial aspects of model-measurement statistics

Scatter plots of the individual-site model versus measurement 10-y means of $NO_2$_daymean, $O_3$_max8hmean, $PM_{10}$_daymean, and 2-y means for $PM_{2.5}$_daymean, by site type, are shown in Figure 1 and illustrate the extent of model-measurement spatial correlation across the UK. The data in these plots are additionally categorised according to the latitude of the monitor site. The numerical values of model-measurement correlation, FAC2, NMB and MB associated with each plot in Figure 1 are presented in Table 1. The correlation between the normalised bias and the latitude across all sites in a given panel of Figure 1 are given
in Table 2. This table also presents the correlation between normalised bias and modelled 10-y mean temperature by site type and pollutant. The equivalent of Figure 1 with data categorised by mean temperature is shown in SI Figure S1.





### 3.1.1. $NO_2$

Figure 1a shows excellent model-measurement agreement in 10-y mean $NO_2$ across RB sites (spatial correlation coefficient of 0.98, regression slope and intercept of 1.10 and 0.0045 µg m$^{-3}$, $n = 7$). This is further emphasised by the low bias for 10-y

mean $NO_2$ at these 7 RB sites: MB = 0.7 µg m$^{-3}$, NMB = 0.06 and FAC2 = 1.00 (Table 1). Spatial correlation between modelled and measured 10-y mean $NO_2$ was also high at UB sites ($r = 0.68$, $n = 37$) and at UT sites ($r = 0.79$, $n = 16$) (Figure 1a), although modelled $NO_2$ concentrations were, on average, lower than measured concentrations at both types of urban sites. The model-measurement discrepancy was less across all statistics at UB sites (MB = −9.5 µg m$^{-3}$, NMB = −0.31, FAC2 = 0.84) than at UT sites (MB = −34.2 µg m$^{-3}$, NMB = −0.64, FAC2 = 0.13) (Table 1). The negative model bias at urban sites can be

attributed to either or both underestimation of $NO_x$ emissions and the instantaneous dilution of $NO_x$ emissions into a 5 km × 5 km model grid cell irrespective of where the monitor is positioned with respect to emissions of $NO_x$ in reality. If air at the urban monitor is more influenced by $NO_x$ emissions than represented by the model grid average then the model value will underestimate the contributions at the monitor from both primary emitted $NO_2$ and secondary $NO_2$ formed by reaction between primary NO and $O_3$. This model grid dilution effect is particularly pronounced for comparison with monitors at UT sites which

are deliberately sited close to strong sources of $NO_x$.

For both types of urban sites, model-measurement agreement was generally better at lower latitude sites, i.e. for sites in the south of the UK compared with sites in the north (Figure 1a). The slight increase in model negative bias for $NO_2$ in the north does not appear to be related to the absolute concentration of $NO_2$ since the differential is similar across a range of $NO_2$

concentrations at sites in the south and north. Normalised bias was significantly positively correlated with temperature (Table 2, SI Figure S1b), i.e. less negative at higher temperature, which is consistent with the smaller negative bias for southern UK, since average temperature decreases with increasing latitude in the UK.

### 3.1.2. $O_3$

Figure 1b shows that the modelled 10-y mean of daily max 8-h mean $O_3$ concentration was greater than measured at all except one site (the coastal RB at Weybourne); but that all modelled and measured 10-y mean $O_3$ concentrations were within a factor of two except at one UT site, London Marylebone Road, which is a kerbside site exposed to very high traffic flows.

As for $NO_2$, the model-measurement statistics for the 10-y mean $O_3$ at RB sites were very good (NMB = 0.08, MB = 5.8 µg

m$^{-3}$, FAC2 = 1.00, $n = 17$) and better than at the UB sites (NMB = 0.27, MB = 15.1 µg m$^{-3}$, FAC2 = 1.00, $n = 30$) (Table 1). The positive model bias for $O_3$ at UB sites is presumably driven by the same issue as the negative model bias for $NO_2$ at the UB sites: the dilution of model $NO_x$ emissions in urban areas into the 5 km × 5 km model grid means that the model insufficiently simulates the reactive removal of $O_3$ by NO close to the urban monitor. Only two UT sites measured $O_3$ so summary model performance statistics for these sites are not illuminating. The large model overestimation of $O_3$ at the London

Marylebone Road UT site is an extreme example of a regional model not being able to simulate the large local $NO_x$ emissions and consequent local $NO_x$-$O_3$ chemistry by the kerbside of this central London street with very heavy traffic.

The lack of model-measurement spatial correlation in 10-y mean $O_3$ concentration across all RB sites ($r = 0.21$, $p = 0.428$, $n = 17$) (Figure 1b) is driven solely by the outlying model-measurement comparison at the Weybourne site, the cause of which is

unknown. When this site is excluded, there is highly significant spatial correlation between model and measurement across all remaining RB sites ($r = 0.81$, $p < 0.001$, $n = 16$) (Table 1). There was also highly significant spatial correlation between





modelled and measured $O_3$ concentration at UB sites ($r = 0.73$, $p < 0.001$, $n = 30$) (Figure 1b, Table 1), although the lower than unity gradient indicates a trend for a less positive bias at higher $O_3$ concentrations. This is again a reflection of the NO + $O_3$ reaction: higher $O_3$ at an UB monitor is likely because the monitor is sited further from immediate sources of primary NO and so less susceptible to the localised (sub-model-grid) effect. Normalised bias in 10-y mean $O_3$ was not correlated with latitude or long-term temperature at either RB or UB sites (Table 2, Figure 1b and SI Figure 1b).

### 3.1.3. PM$_{10}$

The 10-y mean of daily-mean simulations of $PM_{10}$ concentrations were all within a factor of two of the corresponding measurements for all sites (Figure 1c). The 10-y mean $PM_{10}$ concentrations were well modelled at UB sites (NMB = 0.06, MB = 1.26 μg m$^{-3}$, FAC2 = 1.00, $n = 20$) and the spatial correlation across sites, whilst not particularly high, was statistically significant ($r = 0.58$, $p = 0.007$, $n = 20$) (Table 1). Modelled $PM_{10}$ concentrations were higher than measured at RB sites (NMB = 0.39, MB = 6.6 μg m$^{-3}$, FAC2 = 1.00, $n = 4$) (Figure 1c, Table 1) but were also well correlated ($r = 0.91$, $p = 0.092$) despite the small number of comparison sites and small range in 10-y mean $PM_{10}$ values across the RB sites. In contrast, 10-y mean $PM_{10}$ was lower than measured at UT sites (NMB = $-0.25$, MB = $-7.8$ μg m$^{-3}$, FAC2 = 1.00, $n = 5$) (Figure 1c, Table 1) with no evidence of spatial correlation across the sites ($r = 0.40$, $p = 0.502$). The lower modelled values at UT sites is again due to the issue that primary PM emissions associated with traffic and other urban sources close to the UT monitor are in the model diluted and averaged across the 5 km × 5 km grid resolution.

In general there were no strong associations between model-measurement bias for 10-y mean $PM_{10}$ and latitude, although there was significance for smaller bias at UB sites with higher latitude ($r = -0.48$, $p = 0.031$) (Figure 1c, Table 2) and, correspondingly, a tendency for smaller bias in cooler areas ($r = 0.40$, $p = 0.078$) (SI Figure 1c, Table 2).

### 3.1.4. PM$_{2.5}$

Figure 1d shows that all 2-y mean modelled $PM_{2.5}$ concentrations were within a factor of two of the corresponding site measurements, but that at nearly all sites the model yielded lower $PM_{2.5}$ concentrations than were measured. (Even for the shorter time period used for $PM_{2.5}$ comparisons there were only two RB sites with $PM_{2.5}$ monitors so no further comment is made on these data.) The negative bias was smaller at UB sites (NMB = $-0.27$, MB = $-3.5$ μg m$^{-3}$, FAC2 = 1.00, $n = 28$) than at UT sites (NMB = $-0.38$, MB = $-5.5$ μg m$^{-3}$, FAC2 = 1.00, $n = 5$) (Table 1). There was a trend for model underestimation to be greater at sites with higher $PM_{2.5}$ concentrations (Figure 1d). This trend, and the greater underestimation at UT sites, is for the same reason as given above for $PM_{10}$: the inability of the regional model to capture the localisation of urban emissions, particularly close to traffic sources. The lower biases in model simulations of $PM_{10}$ compared with $PM_{2.5}$ is, at least in part, due to a positive model bias in the simulation of the sea salt component of $PM_{coarse}$, which is an important component of background $PM_{coarse}$ in the UK (AQEG, 2005). In contrast to the other sites, there was a positive model bias at the RB site at Auchencorth Moss in Scotland. However, the long-term average concentration of $PM_{2.5}$ at this site is very low (~5 μg m$^{-3}$) and only about half the next lowest measured $PM_{2.5}$ concentration. Accurate measurement of these very low concentrations of $PM_{2.5}$ is a considerable challenge (AQEG, 2012).

Model-measurement spatial correlation of $PM_{2.5}$ across UB sites was moderate but statistically significant ($r = 0.58$, $p = 0.001$, $n = 28$). As with $PM_{10}$, there was no strong association between model bias for $PM_{2.5}$ and geographical location (Table 2, Figure 1d and SI Figure 1d) although there was a tendency for smaller bias with higher latitude ($r = -0.28$, $p = 0.141$) and in





cooler areas ($r = 0.43$, $p = 0.022$). This may indicate a negative bias in simulating secondary PM components that have smaller concentrations in the north of the UK compared with the south which is more influenced by transport of these components and of their precursors from continental Europe (Vieno et al., 2014).

5 ## 3.2. Evaluation of temporal aspects of model-measurement statistics

### 3.2.1. Statistics for daily metrics across the full simulation period

Table 3 summarises the individual-site model vs measurement FAC2, NMB and $r$ statistics, grouped by site type, for the 10 years of daily $NO_2$, $O_3$, $PM_{10}$ concentrations, and 2 years of daily $PM_{2.5}$ concentrations. Statistics for an individual site are derived from up to 3,652 pairs of daily model-measurement data.

The temporal variability in daily $NO_2$ and $O_3$ over the 10 years was well captured by the model at both RB and UB sites. The median (25th percentile, 75th percentile, no. of sites) model-measurement correlation coefficients for $NO_2$_daymean across RB and UB sites were 0.75 (0.73, 0.78, $n = 7$) and 0.70 (0.63, 0.77, $n = 37$), respectively, whilst for $O_3$_max8hmean they were 0.73 (0.72, 0.76, $n = 17$) and 0.76 (0.74, 0.78, $n = 30$), respectively. Model-measurement NMB for $NO_2$ and $O_3$ at RB sites 15 was also small. The median (25th percentile, 75th percentile) NMB across RB sites for the 10 years of $NO_2$_daymean and $O_3$_max8hmean were 0.08 (0.02, 0.12) and 0.11 (0.08, 0.12), respectively. The corresponding NMB data across UB sites were larger, −0.29 (−0.40, −0.12) and 0.26 (0.18, 0.32) for $NO_2$_daymean and $O_3$_max8hmean respectively, with the explanations for the negative and positive bias values for $NO_2$ and $O_3$, respectively, at urban locations as described above.

20 Table 3 shows that the agreement between modelled and measured temporal variability in daily $PM_{2.5}$ over the 2 years of available data was also reasonable. The median (25th percentile, 75th percentile, no. of sites) model-measurement temporal correlation coefficients for $PM_{2.5}$_daymean across RB and UB sites were 0.65 (0.64, 0.65, $n = 2$) and 0.69 (0.67, 0.73, $n = 28$), respectively. The correlations for $PM_{10}$_daymean were poorer, with corresponding data for correlation coefficients across RB and UB sites for the 10 years of available data of 0.47 (0.46, 0.48, $n = 4$) and 0.50 (0.45, 0.55, $n = 20$). However, although 25 temporal correlation was acceptable for $PM_{2.5}$_daymean there was substantial bias, with median (25th percentile, 75th percentile) NMB values at RB and UB sites of 0.38 (0.18, 0.59) and −0.26 (−0.33, −0.22), respectively (but note only two sites featured in the RB comparison).

### 3.2.2. $NO_2$_daymean grouped by different periods of time

30 Figure 2 shows box-whisker plots summarising the individual site model-measurement FAC2, NMB and $r$ statistics for daily mean $NO_2$, with the daily data grouped by year, by month, and by day of week. All box plots indicate substantial inter-site variability in model-measurement statistics, but also differences in these statistics between site type and, in some instances, between the individual blocks of time over which the data are averaged.

35 *By year.* Figure 2a shows there were no long-term trends in the model-measurement correlations of daily mean $NO_2$ across the years, for rural or for urban sites. At RB sites, a high fraction of modelled daily mean $NO_2$ was within a factor of two of the measurements, without inter-annual trend (10-y mean of the median FAC2 each year = 0.85) (Figure 2b). There was some inter-year variation in the model-measurement NMB at RB sites which, although near zero on average for years 2001-2003 and 2007-10 (mean of median NMB = 0.03) was positive in years 2004-2006 (mean of median NMB = 0.18) (Figure 2c). The 40 model accuracy at both types of urban sites showed a slight trend to lower FAC2 (Figure 2b) and greater negative NMB (Figure





2c) in years 2008-2010. The larger model-measurement bias in the latter, whilst similar values of correlation are retained, is potentially indicative of shortcomings in emissions totals in these latter years of the study.

*By month.* The model-measurement statistics for daily mean $NO_2$ exhibited some seasonal variability (Figure 2d-f). Figure 2d
shows that there was a similar small seasonal variation in model-measurement correlation at all site types, with higher correlation coefficients on average in autumn and winter, and lower correlation coefficients in spring and summer. Correlation was fairly similar between site types, better on average for RB and UB sites and slightly poorer at UT sites. Model bias was smallest at RB sites, and whilst FAC2 at RB sites was fairly constant between months (Figure 2e), the median NMB at RB sites varied between a median of $-0.07$ in March and a median of $0.21$ in October (Figure 2f). In contrast, in urban areas,
model-measurement difference was least in winter months, December-January-February (mean of median FAC2 = 0.72 and 0.28, mean of median NMB = $-0.28$ and $-0.59$, for UB and UT sites, respectively), and lowest in late spring and early summer (mean of median FAC2 = 0.67 and 0.06, mean of median NMB = $-0.33$ and $-0.73$, over May, June and July for UB and UT sites, respectively) (Figures 2e and 2f).

These seasonal variations may have a variety of causes. In terms of chemical and meteorological effects, the $NO + O_3$ titration effect already described will be greater in summer than in winter, and the model grid dilution effect will be exacerbated in summer by greater convective boundary-layer mixing. Some part of the explanation for poorer model-measurement accuracy in summary may also be due to shortcomings in the values of the monthly emission factors used in the model to disaggregate the annual emissions totals of $NO_x$ (and VOC). The more consistent temporal correlations across site types compared with bias
is again consistent with issues with the specification of amount and dilution of local emissions into the 5 km model grids rather than issues with describing the meteorology.

*By day of week.* Model-measurement correlation for daily mean $NO_2$ was similar for all days of the week at all site types (Figure 2g). On the other hand, there were pronounced differences in NMB between weekday and weekend for both RB and
UB sites (Figure 2i). NMB was more positive at weekends at RB sites than during weekdays, and NMB was similarly less negative at weekends compared with weekdays. The invariant day-of-week correlation but weekday/weekend differences in NMB again indicates that general meteorology is captured well by the model but that there may be shortcomings in the day-of-the-week factors applied in the model to disaggregate the annual local $NO_x$ (and VOC) emission totals.

**3.2.3. $O_3$_max8hmean grouped by different periods of time**

As with daily mean $NO_2$, Figure 3 reveals some trends in model-measurement statistics for daily maximum 8-h mean $O_3$ for data grouped by year, month, and day of week. There are only two UT sites for $O_3$ comparisons, and one of these is the 'extreme' kerbside site of London Marylebone Road, so data for UT sites are not discussed further.

*By year.* Figures 3a-c show that the $O_3$_max8hmean model-measurement statistics at RB and UB sites remained fairly constant over the years 2001-2010. Model-measurement correlations were similar at both types of sites (mean of median $r$ = 0.76 and 0.77 for RB and UB sites, respectively) (Figure 3a), but bias was less at RB than at UB sites (mean of median FAC2 = 0.98 and 0.87, mean of median NMB = 0.10 and 0.33, respectively) (Figures 3b and 3c).

*By Month.* Model-measurement correlation exhibited a pronounced seasonal variation (but which was similar for both RB and UB sites), with much better correlation in winter and summer than in spring and autumn (Figure 3d). On the other hand, model bias was generally lower in spring and summer than in autumn and winter, with the smallest bias in June, and the greatest in



October (Figure 3f). This seasonal variation in bias was more pronounced at UB sites than at RB sites. As discussed above for NO$_2$, the seasonal trends in O$_3$ model biases may be due to shortcomings in assigning seasonal trends to emissions of NO$_x$ and reactive VOC that together impact on regional O$_3$ concentrations. However, many factors influence surface concentrations of O$_3$, acting on different temporal and spatial scales (Royal Society, 2008), so the seasonal patterns in correlation and bias are likely the net consequence of a number of drivers.

*By day of week.* Model-measurement correlation at both types of background sites did not show variation with day of the week (mean of median $r$ = 0.74 and 0.76 for RB and UB sites, respectively) (Figure 3g). Correlation was much poorer at the Weybourne RB site ($r$ = ~0.29), but, as noted above, the Weybourne comparison (which is only for O$_3$) is clearly anomalous. Model-measurement bias at RB sites was largely similar across day-of-week (mean of median FAC2 = 0.97, mean of median NMB = 0.11), with slightly reduced positive bias on weekend days (Figures 3h and 3i). At UB sites, bias was greater during Tuesday-Friday (mean of median NMB = 0.30 and mean of median FAC2 = 0.87), but mean NMB reduced to 0.15 on Sundays and mean FAC2 increased to 0.95 (Figures 3h and 3i). The positive model bias at the urban sites, plus the improved model bias over the weekend, both indicate the issue of dilution into the 5 km × 5 km model grid of urban NO$_x$ emissions and the consequent lack of capture of the NO reaction with O$_3$ at sites influenced by traffic emissions (which are lower in the model at weekends).

### 3.2.4. PM$_{10}$_daymean grouped by different periods of time

*By year.* Model-measurement correlations of daily mean PM$_{10}$, grouped by year, did not show any inter-annual trend across the 10-y evaluation period or across the three site types (Figure 4a), except for enhanced correlations, on average, in 2003. Annual averages of model-measurement accuracy in daily PM$_{10}$ showed some inter-annual variabilities (Figures 4b and 4c for FAC2 and NMB) but no trends across the 10 years.

*By month.* Model-measurement comparison statistics for daily mean PM$_{10}$ displayed strong seasonality at all three types of sites (Figure 4d-f). Correlations were similar for the three types of site, with the best correlation in summer and the worst in late autumn and winter (Figure 4d). In terms of bias, at RB sites PM$_{10}$ concentration was best simulated in late summer (mean of median NMB = 0.04 for July and August), and most overestimated in late autumn (NMB = 0.69 for October) (Figure 4f). A similar seasonal pattern was apparent at the urban sites, but superimposed on a more general negative bias. Thus, at UB sites, PM$_{10}$ concentration was underestimated in late summer, but overestimated in late autumn and winter, with better accuracy on average in the summer half of the year. At UT sites, negative bias for PM$_{10}$ concentration was greatest in summer (mean of median NMB = −0.42 for July and August) and least in late autumn (NMB = −0.13 for October).

*By day of week.* Patterns in day-of-week model-measurement statistics for daily mean PM$_{10}$ (Figure 4g-i) showed some similarity with those for daily mean NO$_2$ (Figure 2g-i). Model-measurement correlations were fairly consistent throughout the week and similar at all site types (Figure 4g) (a small reduction in correlation on Wednesdays at RB sites is likely simply a statistical artefact. There was no significant variation in model accuracy at RB with day of the week (Figures 4h and 4i), although there are only 4 sites for this comparison. At UB sites, PM$_{10}$ concentration was simulated most accurately on weekdays (mean of median NMB = 0.01, mean of median FAC2 = 0.87) (Figures 4h and 4i), but was overestimated at RB sites (mean of median NMB = 0.41) and was underestimated at UT sites (mean of median NMB = −0.25). The positive bias at RB sites was probably due to the overestimation of sea salt, as mentioned above, and the underestimation at UT sites could be attributed to the dilution and underestimation of local primary PM$_{10}$ from traffic sources, e.g., from tyre/brake wear. At weekends, positive bias in PM$_{10}$ concentrations increased at UB sites, whereas the negative bias at UT sites reduced, suggesting





that the day-of-week emission factors used in the model might not adequately reflect actual weekday-weekend differences in emissions.

Again, the general consistency in temporal correlation with site type and time period, compared with the variation in bias, is consistent with the main driver of model shortcoming being in accuracy of emissions (totals and temporal disaggregation) rather than in simulation of atmospheric chemistry and transport processes.

### 3.2.5. PM$_{2.5}$_daymean grouped by different periods of time

*By year.* Figures 5a-c summarise the model evaluation statistics for PM$_{2.5}$ daily means for the 2-y period of available monitor data (2009-10). The PM$_{2.5}$ model-measurement comparison statistics are generally poorer in 2010 but two years is insufficient to draw any conclusion on inter-annual trend As for PM$_{10}$ daily mean comparisons, there was positive bias for daily mean at RB sites (mean of median NMB = 0.39) and negative bias at UB and UT sites (mean of median NMB = −0.26 and −0.41 at UB and UT sites, respectively) (Figure 5c). However, PM$_{2.5}$ was measured at only two RB sites, and at one of these, Auchencorth Moss in Scotland, the PM$_{2.5}$ concentrations were substantially lower than at any of the other measurement sites. At least half of the modelled PM$_{2.5}$ daily mean concentrations were within a factor of two of the measurements at all sites, except the RB site of Auchencorth Moss and the UT site of Bury Roadside (Figure 5b). Of the two RB sites, the model accurately simulated daily mean PM$_{2.5}$ concentration at Harwell (mean NMB = −0.02, mean FAC2 = 0.90), but there was substantially positive bias at Auchencorth Moss (mean NMB = 0.81, FAC2 = 0.43).

*By month.* Model-measurement correlation was generally better in the summer half of the year than in the winter half (e.g. mean of median $r$ = 0.76 and 0.68, respectively, at UB sites) (Figure 5d). Similarly, there were greater values of FAC2 in spring and summer than in autumn and winter, particularly at UB sites (mean of median FAC2 = 0.86 and 0.78, respectively) (Figure 5e). On the other hand, model-measurement bias did not vary with season (Figure 5f).

*By day of week.* In contrast to the other three pollutants, there was no obvious differences in model-measurement statistics between weekdays and weekend at any of the three types of site (Figure 5g-i), but there are substantially less comparison data for PM$_{2.5}$ than for the other three pollutants.

### 3.2.6. Hourly model-measurement statistics

The focus in this work was model-measurement comparisons at daily and annual averaging resolution, but concentration data were available at hourly resolution and the Supplementary Information presents figures and discussion of the comparison statistics for NO$_2$ and O$_3$ averaged by hour of day. These data support the general observations presented above for the longer averaging periods, in particular that correlations between model and measurement hourly data were generally consistent throughout the day but that bias showed systematic variation, which is interpreted as error in the hour-of-day emissions factors used to disaggregate the annual NO$_x$ emissions totals in the model (and to over-dilution of the NO$_x$ emissions into the model grid compared to the siting of the monitor, particularly for UT sites).





## 4. Discussion

The work presented here was motivated by the use of the EMEP4UK-WRF model output for air pollution epidemiology and health burden assessment; therefore the model-measurement comparison focused on health-relevant metrics for the most important ambient air pollutants: specifically the annual and daily means for $PM_{10}$, $PM_{2.5}$, $NO_2$ and $O_3$ (the daily maximum 8-h mean for $O_3$) (WHO, 2013a). The model-measurement comparison was comprehensive; all available data from all monitors in the UK's national automated urban and rural network for 2001-2010 were used, which span the range of ambient environments in which people are exposed to air pollution in the UK. Focus was placed on the two most important statistics highlighted in the literature for evaluation of air quality model output against policy (and hence health) relevant standards: correlation (temporal and spatial) and bias (e.g. USEPA, 2007; Derwent et al., 2010; Thunis et al., 2012).

Even for a well-specified Eulerian model (in terms of input data, transport, chemistry, etc.), model-measurement agreement may not be perfect for (at least) the following two reasons: first, the model simulates a volume-averaged concentration whereas the monitor records the composition of the air in one part of that volume, which may or may not reflect the average concentration for the whole volume over the relevant time-averaging period; and, secondly, the measurement may be in error. A rural background monitor in homogenous terrain and well-away from local sources may be anticipated to be sampling air that is more homogenous over the 5 km × 5 km model grid in which it is located than an urban traffic monitor that is deliberately sited close to a major source of air pollutant emissions and therefore not representative of the composition of the atmosphere averaged over the model grid. The representativeness of an urban background monitor for the air in the 5 km × 5 km model grid in which it is located will be between these two extremes and to some extent dependent on the size of the urban area, as well as the distance of the monitor from specific local pollutant emission sources.

The presence of measurement uncertainty degrades the values that can be expected from air quality model-measurement statistics. Thunis et al. (2012) developed a series of relationships that define minimum values for model-measurement statistics, given a value, $U$, for measurement uncertainty; for example, $|NMB| < 2U/\bar{O}$ and $r > 1 - 2(U/\sigma_O)^2$. They then estimated minimum values for these statistics by taking example values for $\bar{O}$ and $\sigma_O$ from more than 700 monitoring stations around Europe (for 2009) and using the measurement data quality objectives for measurement uncertainty specified in the EU Air Quality Directive as values for $U$. For daily maximum 8-h mean $O_3$ and daily mean $PM_{10}$ these are 15% and 25%, respectively (EC Directive, 2008). At these levels of measurement uncertainty, model-measurement correlation coefficients for daily mean $PM_{10}$ as low as 0.40-0.48 (the range reflects the three different types of measurement site) still satisfy the model-measurement performance criterion (Thunis et al., 2012). For daily maximum 8-h mean $O_3$ the minimum values for $r$ to satisfy the criterion are in the range 0.54-0.69. Minimum values for $|NMB|$ for daily mean $PM_{10}$ are in the range 0.57 to 0.58, and for daily maximum 8-h mean $O_3$ are in the range 0.32 to 0.33 (Thunis et al., 2012). Values of these statistics for daily mean $PM_{2.5}$ and daily mean $NO_2$ are anticipated to be similar to those above for $PM_{10}$ and $O_3$, respectively. The above values are presented in Table 3 for comparison against the $r$ and NMB values derived in the present model-measurement comparison. If measurement uncertainty is greater than specified in the data quality objectives, for example for measurement of concentrations lower than the relevant air quality limit value, as the majority of concentrations are, then lower values of $r$, and greater values of $|NMB|$, than quoted above define satisfactory model-measurement comparison (Thunis et al., 2013; Pernigotti et al., 2013).

Table 3 shows that in the large majority of instances the values of model-measurement correlation and NMB from this EMEP4UK-WRF modelling exceed the threshold values described above for satisfactory model performance in the presence of measurement uncertainties at the levels assigned. For example, the 25th percentile across sites of EMEP4UK-WRF model-measurement correlation for daily maximum 8-h mean $O_3$ at RB and UB sites ($r = 0.72$ and 0.74, respectively) well exceed the values of 0.54 and 0.69 derived by Thunis et al. (2012). Likewise, the 75th percentile of EMEP4UK-WRF model-



measurement NMB values for the $O_3$ metric (0.12 and 0.32 for RB and UB sites) are lower than the respective Thunis et al. (2012) values of 0.32 and 0.33. The EMEP4UK-WRF model-measurement statistics for $O_3$ at the two UT sites are, however, poorer (Table 3). For, daily mean $PM_{10}$ the 25th percentile values of EMEP4UK-WRF model-measurement correlation coefficients are very similar to those of Thunis et al. (2012), but EMEP4UK-WRF model-measurement NMB values are

generally much lower than those of Thunis et al. (2012). The situation is similar (better for correlation) for $PM_{2.5}$, when assigning the $PM_{10}$ satisfactory performance values to $PM_{2.5}$ also (Table 3). As described in Section 2, instrumentation for 'real time' measurement of $PM_{10}$ and $PM_{2.5}$ has varied and in some instances has necessitated post hoc application of correction factors, which increases measurement uncertainty for these species compared with measurement of $NO_2$ and $O_3$.

The UK AURN operates as a single network subject to standardised QA/QC procedures (as described in the Section 2) so measurement uncertainty might be expected to be lower than the values used by Thunis et al. (2012). On the other hand, this analysis of magnitudes of model-measurement statistics does not allow for uncertainty arising from lack of spatial representativeness of the measurement location within its model grid, as discussed already.

Although the model-measurement statistics reported in this work are for the most part in line with or better than expectations, there were also instances of trends in statistics with site type, month-of-year and day-of-week. (In general there were no obvious inter-annual trend across the decade of comparisons.) There was generally less bias at the background sites compared with traffic sites, and bias was least overall for rural background sites (e.g. median normalised mean bias values for $O_3$ and $NO_2$ of 0.08 and 0.11, respectively), reflecting the smaller likelihood for sub-grid variations in sources, dispersion and deposition to

perturb concentrations at the monitor location away from the model grid average. There was a tendency for positive model bias for $O_3$ at UB sites (median NMB = 0.26) and for negative model bias in $NO_2$ (−0.29) and $PM_{2.5}$ (−0.26) at these sites. The negative biases may reflect both underestimation of primary emissions of $NO_x$ and PM and a tendency for air at urban background monitor locations to be more influenced by the primary emissions in the vicinity than simulated by the model which effectively averages all emissions evenly across the 5 km × 5 km grid in which the monitor is located. Unless the urban

area is very large – greater than a few km in linear dimension – then the air even at a background site in the centre of that urban area is likely to be more influenced by local primary emissions than peripheral (suburban) parts of the urban area included in the model grid average. A further contributor to model negative bias for PM are known omissions in the model of some PM components, including particle-bound water and some sources of dust resuspension.

The positive model bias for $O_3$ at UB sites is consistent with the explanations given above for the negative model biases for $NO_2$ (and $PM_{2.5}$). The dilution of the $NO_x$ emissions in urban areas into the 5 km × 5 km model grid means that the model underestimates the reactive removal of $O_3$ by NO in the vicinity of the urban monitor. These sub-grid effects are particularly acute for roadside and kerbside sites which are deliberately sited close to strong sources of $NO_x$ and PM, and which cannot be resolved even by the comparatively high resolution of the EMEP4UK-WRF ACTM.

Instances of trends in model-measurement bias with month or day of the week are described in the Results section. The generally good daily temporal correlations discussed already indicate that the model captured the day-to-day changes in air mass movements which are the strongest influences on surface concentrations of pollutants at this temporal resolution. The observed seasonal and weekday/weekend variations in bias (and of diurnal variations in bias – see Supplementary Information)

are therefore strongly suggestive of shortcomings in the monthly and weekday/weekend (and hour-of-day) emissions factors applied in the model to disaggregate the annual total emissions supplied by the emissions inventories.





As stated at the outset, the motivation here was use of the EMEP4UK-WRF model output for health studies. In the context of use of concentration data for epidemiology, in the broadest terms correlation is more important than bias, and for the model output reported here, model-measurement correlations (both temporal and spatial) were generally considerably better, particularly for the gaseous pollutants, than bias statistics. Epidemiological studies of association of ambient air pollution with health require an estimate of exposure for each subject, most usually from measurements from monitors but increasingly from models. The difference between the estimates and a hypothetical gold standard, for example concentration outside the residence of each subject, is called exposure measurement error. (It is assumed here that it is the association of ambient pollution with health outcome at the small-area level that is important, because of the link to regulation (Dominici et al., 2000), rather than exposure at the level of the individual, and therefore issues of disparity between the concentration at a location and true personal exposure are not considered.) The consequences of measurement error are to reduce the power of the study to detect an association and to bias the magnitude of the association (Sheppard et al., 2005; Sheppard et al., 2012; Armstrong and Basagaña, 2015).

The agreement statistics determining the magnitude of this 'blunting' depends on the specific context. Study power is simplest, depending only on the correlation between the true and estimated exposure. Of the two main types of epidemiological studies of air pollution: in 'spatial studies' power is diminished according to the correlation of long-term true and estimated means over space; in 'time series studies' it depends on correlations of daily values over space. Thus the model-measurement correlations reported in Sections 3.1 and 3.2 have a fairly direct implication for study power in those two study types except that errors in the measured values as estimates of the mean over the population in the grid square (or wider area) are not allowed for. Because of this, the power of studies using modelled concentrations would be somewhat better than implied by the correlations reported (Butland et al., 2013).

Low correlation of 'true' and estimated exposures also often reduces estimated size of association (e.g. relative risk per unit exposure), but other aspects of the error distribution also matter, notably the extent of Berkson or classical type (Butland et al., 2013; Armstrong and Basagaña, 2015). It is difficult and beyond the scope of this paper, to separate Berkson and classical error, but in the absence of this it would be reasonable to consider the model-measured correlations as broad guides to bias in association as well as power. Perhaps surprisingly, additive bias (e.g. estimating concentration 10 units too high on average) has little effect in epidemiological studies, at least if the exposure-health association is assumed linear, as it usually is (although bias in association is also dependent on relative magnitudes of variance in 'true' and estimated exposures).

As well as the good temporal correlations for daily pollutant metrics, the good spatial correlations between long-term averaged modelled and measured concentrations across UB sites for all four pollutants selected encouragingly suggests that the EMEP4UK-WRF modelled pollutant concentration may broadly reduce exposure measurement error caused by using pollution measurements from air pollution monitors far from the population under consideration. On the other hand, a bias error in the simulations contributes to uncertainty in the investigation of any threshold in concentration-health effect, and in health impact assessments that apply concentration-response functions to estimated concentrations of exposure.

This study has worked with the EMEP4UK-WRF v4.3 model. Model-measurement statistics will be different for other models. However, other ACTM are similarly constructed and so the broad discussion points relating to intrinsic limitations to monitor versus grid-volume comparison statistics, unresolved sub-grid variabilities, and shortcomings in magnitudes and temporal trends in emissions are generalizable. Local dispersion models can better represent the sources and dispersion at high spatial resolution but these can only be configured for specific urban areas at a time, are similarly constrained by the accuracy of the spatiotemporal emissions data and require provision of boundary conditions of meteorology and atmospheric composition





(often supplied by an ACTM). Dispersion models have also been combined with land-use regression models (Wilton et al., 2010; Michanowicz et al., 2016) but again for individual areas only. Some progress is being made in combining measurement (both ground-based and satellite) and model data through data assimilation (e.g. (MACC-II: Monitoring Atmospheric Composition and Climate - Interim Implementation (www.gmes-atmosphere.eu/about/); Singh et al., 2011) and data fusion

5 (Berrocal et al., 2011; Zidek et al., 2012; Friberg et al., 2016), but these approaches are computationally demanding, particularly for reactive species, and can only be applied to historic data. National-scale air pollution modelling as described here, despite acknowledged limitations for health studies (Butland et al., 2013), has the benefit of providing self-consistent chemical concentration fields, data for air pollutant components that are either not, or only sparsely, measured and provide the capacity to investigate the potential effects of alternative possible futures.

## 5. Conclusions

This study was motivated by the use in air pollution epidemiology and health burden assessment of data simulated at 5 km × 5 km horizontal resolution by the EMEP4UK-WRF v4.3 atmospheric chemistry transport model. A spatially and temporally comprehensive set of model-measurement comparison statistics are presented for daily and annual concentrations of $NO_2$, $O_3$,

$PM_{10}$ and $PM_{2.5}$ across the UK for a 10 year period.

In general for epidemiology, capturing correlation is more important than bias, and in this study model-measurement temporal correlation of daily concentrations was generally better than expectations reported in the literature that take into account potential measurement uncertainties. Model-measurement bias varied according to monitor site classification with generally

less bias at the rural and urban background sites compared with urban traffic sites. Bias was least overall for rural background sites. The greater consistency in temporal correlation with site type and across months and day of week, compared with variations in bias, is strongly indicative that the main driver of model shortcoming is inaccuracy of emissions (totals and the monthly and day-of-week temporal factors applied in the model to the totals) rather than in simulation of atmospheric chemistry and transport processes.

Despite discussed limitations, these detailed analyses support use of model data such as these in air pollution epidemiology. Air pollution modelling at the spatial coverage and spatial resolution described here has the benefit of increasing study power, of providing data for air pollutant components that are either not, or only sparsely, measured and of enabling investigation of the potential effects of alternative future scenarios.

## Code and data availability

This study used output from the EMEP4UK-WRF model which is a regional application of the European Monitoring and Evaluation Programme (EMEP) MSC-W model (available at www.emep.int, version vn4.3 used here) driven by meteorology

from the Weather Research and Forecast model (www.wrf-model.org) version 3.1.1. As described and referenced in Section 2.1, the EMEP4UK model has increased spatial resolution over a British Isles inner domain and uses national emissions data for the UK. All EMEP4UK modifications are included in the official EMEP model. The model output described here is archived at the University of Edinburgh and available on request

**Competing Interest**

The authors declare that they have no conflict of interest.



**Acknowledgements**

This work was supported by funding from the Natural Environment Research Council and Medical Research Council Environmental Exposure and Human Health Initiative (EEHI) grants NE/I007865/1, NE/I007938/1 and NE/I008063/1. The

5   EMEP4UK model is also supported by the UK Department for the Environment, Food and Rural Affairs (Defra) and the NERC Centre for Ecology & Hydrology (CEH). We also acknowledge access to the AURN measurement data, which were obtained from uk-air.defra.gov.uk and are subject to Crown 2014 copyright, Defra, licenced under the Open Government Licence (OGL).



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





**Table 1:** Numbers of AURN sites satisfying the data capture criteria described in the text, and model-measurement statistics for the 10-y means of NO$_2$_daymean, O$_3$_max8hmean, PM$_{10}$_daymean, and for the 2-y means of PM$_{2.5}$_daymean. The latter data provide a measure of the spatial agreement between modelled and measured pollutant concentrations across the UK.

| | *n* | *r* | FAC2 | NMB | MB / µg m$^{-3}$ |
|---|---|---|---|---|---|
| **NO$_2$_daymean (2001-2010)** | | | | | |
| Rural Background | 7 | 0.98 | 1.00 | 0.06 | 0.68 |
| Urban Background | 37 | 0.68 | 0.84 | −0.31 | −9.52 |
| Urban Traffic | 16 | 0.79 | 0.13 | −0.64 | −34.16 |
| | | | | | |
| **O$_3$_max8hmean (2001-2010)** | | | | | |
| Rural Background | 17 | 0.21 (0.81[a]) | 1.00 | 0.08 | 5.80 |
| Urban Background | 30 | 0.73 | 1.00 | 0.27 | 15.08 |
| Urban Traffic | 2 | 1.00 | 0.50 | 0.78 | 30.70 |
| | | | | | |
| **PM$_{10}$_daymean (2001-2010)** | | | | | |
| Rural Background | 4 | 0.91 | 1.00 | 0.39 | 6.56 |
| Urban Background | 20 | 0.58 | 1.00 | 0.06 | 1.26 |
| Urban Traffic | 5 | 0.40 | 1.00 | −0.25 | −7.79 |
| | | | | | |
| **PM$_{2.5}$_daymean (2009-2010)** | | | | | |
| Rural Background | 2 | 1.00 | 1.00 | 0.19 | 1.32 |
| Urban Background | 28 | 0.58 | 1.00 | −0.27 | −3.51 |
| Urban Traffic | 5 | 0.49 | 1.00 | −0.38 | −5.47 |

[a] Value of *r* when the outlier site for RB O$_3$ measurements (Weybourne) is discounted.

**Table 2:** Correlation of the normalised bias between model and measurement 10-y means of pollutant daily metrics (2-y mean for PM$_{2.5}$) at a site with the latitude or with the 10-y mean temperature at that site. Correlations significant at *p* <0.05 are highlighted in bold. RB, rural background; UB, urban background; UT, urban traffic.

| Pollutant | *n* | Correlation between normalised bias and stated variable | |
|---|---|---|---|
| | | **Latitude** | **Temperature** |
| NO$_2$ (RB) | 7 | 0.20 (*p* = 0.671) | −0.16 (*p* = 0.730) |
| NO$_2$ (UB) | 37 | **−0.53** (*p* < 0.001) | **0.37** (*p* = 0.026) |
| NO$_2$ (UT) | 16 | −0.48 (*p* = 0.058) | **0.51** (*p* = 0.045) |
| | | | |
| O$_3$ (RB) | 17 | 0.24 (*p* = 0.353) | −0.39 (*p* = 0.119) |
| O$_3$ (UB) | 30 | 0.12 (*p* = 0.530) | −0.08 (*p* = 0.674) |
| | | | |
| PM$_{10}$ (RB) | 4 | 0.66 (*p* = 0.340) | −0.68 (*p* = 0.324) |
| PM$_{10}$ (UB) | 20 | **−0.48** (*p* = 0.031) | 0.40 (*p* = 0.078) |
| PM$_{10}$ (UT) | 5 | −0.35 (*p* = 0.558) | 0.38 (*p* = 0.532) |
| | | | |
| PM$_{2.5}$ (UB) | 28 | −0.28 (*p* = 0.141) | **0.43** (*p* = 0.022) |
| PM$_{2.5}$ (UT) | 5 | 0.25 (*p* = 0.681) | -0.42 (*p* = 0.481) |





**Table 3:** Median (25[th] percentile, 75[th] percentile) values of the *n* individual-site model-measurement statistics of daily pollutant metric for the full 10-y period (2-y period for PM$_{2.5}$), grouped by site type: RB, rural background; UB, urban background; UT, urban traffic. Also shown are the minimum values for *r* and |NMB| presented by Thunis et al. (2012) for satisfactory model-measurement comparisons for the given air quality metric assuming there is uncertainty in the measurement at the maximum allowed measurement uncertainties of 15% for the O$_3$ metric and 25% for the PM$_{10}$ metric specified in the EU air quality directive. The minimum values of *r* and |NMB| derived for the O$_3$ and PM$_{10}$ metrics are assigned to the NO$_2$ and PM$_{2.5}$ metrics, respectively, and distinguished by putting in italics. See text for further details.

| | *n* | *r* | FAC2 | NMB | MB / µg m$^{-3}$ | Min MPC[a] | |
|---|---|---|---|---|---|---|---|
| | | | | | | *r* | \|NMB\| |
| **NO$_2$_daymean** | | | | | | | |
| **RB** | 7 | 0.75 (0.73, 0.78) | 0.86 (0.82, 0.87) | 0.08 (0.02, 0.12) | 0.94 (0.35, 1.31) | *0.54* | *0.32* |
| **UB** | 37 | 0.70 (0.63, 0.77) | 0.73 (0.61, 0.88) | −0.29 (−0.40, −0.15) | −9.18 (−14.60, −3.22) | *0.69* | *0.33* |
| **UT** | 16 | 0.55 (0.44, 0.62) | 0.18 (0.09, 0.31) | −0.66 (−0.74, −0.57) | −31.61 (−43.42, −25.64) | *0.68* | *0.33* |
| | | | | | | | |
| **O$_3$_max8hmean** | | | | | | | |
| **RB** | 17 | 0.73 (0.72, 0.76) | 0.97 (0.96, 0.99) | 0.11 (0.08, 0.12) | 7.22 (5.66, 8.00) | 0.54 | 0.32 |
| **UB** | 30 | 0.76 (0.74, 0.78) | 0.89 (0.85, 0.94) | 0.26 (0.18, 0.32) | 14.30 (11.10, 17.87) | 0.69 | 0.33 |
| **UT** | 2 | 0.58 (0.57, 0.60) | 0.56 (0.45, 0.68) | 0.95 (0.70, 1.19) | 30.70 (27.74, 33.66) | 0.68 | 0.33 |
| | | | | | | | |
| **PM$_{10}$_daymean** | | | | | | | |
| **RB** | 4 | 0.47 (0.46, 0.48) | 0.75 (0.69, 0.82) | 0.43 (0.26, 0.59) | 6.17 (5.13, 7.60) | 0.48 | 0.58 |
| **UB** | 20 | 0.50 (0.45, 0.55) | 0.86 (0.84, 0.88) | 0.03 (−0.01, 0.14) | 0.61 (−0.20, 2.69) | 0.44 | 0.58 |
| **UT** | 5 | 0.45 (0.40, 0.53) | 0.77 (0.63, 0.80) | −0.22 (−0.33, −0.21) | −7.12 (−9.85, −5.97) | 0.40 | 0.57 |
| | | | | | | | |
| **PM$_{2.5}$_daymean** | | | | | | | |
| **RB** | 2 | 0.65 (0.64, 0.65) | 0.66 (0.55, 0.78) | 0.38 (0.18, 0.59) | 1.32 (0.54, 2.09) | *0.48* | *0.58* |
| **UB** | 28 | 0.69 (0.67, 0.73) | 0.81 (0.76, 0.85) | −0.26 (−0.33, −0.22) | −3.43 (−4.74, −2.91) | *0.44* | *0.58* |
| **UT** | 5 | 0.73 (0.65, 0.75) | 0.58 (0.56, 0.69) | −0.41 (−0.45, −0.37) | −6.12 (−6.68, −6.00) | *0.40* | *0.57* |

[a] Minimum model performance criteria. See Thunis et al. (2012) for details on the derivation of the criteria and the estimation of the values for these air pollutant metrics.

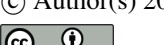



**Figure 1**: Scatter plots of the 10-year means of the modelled and measured pollutant daily metrics at each site, grouped by site type, and with data markers shaded according to the latitude of the measurement site: (a) $NO_2$; (b) $O_3$; (c) $PM_{10}$; (d) $PM_{2.5}$. The solid and dashed lines are the 1:1, and the 2:1 and 1:2 lines, respectively. The values of $r$, FAC2 and NMB associated with the data in each plot are given in Table 1.







**Figure 2:** Model-measurement statistics per site for NO$_2$ daily mean concentrations during 2001-2010, by site type, and by (a-c) year, (d-f) month of year, and (g-i) day of week. (a), (d) and (g) are Pearson's correlation coefficient ($r$); (b), (e) and (h) are fraction of data pairs within a factor of two (FAC2); and (c), (f) and (i) are normalised mean bias (NMB). Dots show individual site statistics ($n = 7$, 37 and 16 for RB, UB and UT sites respectively), which are summarised in the superimposed box-plot whose shading demarcates the interquartile range (IQR) and whose whiskers extend to the largest and smallest value within 1.58 × IQR from the box hinges.




**Figure 3:** Model-measurement statistics per site for O$_3$ daily maximum 8-h mean concentrations during 2001-2010, by site type, and by (a-c) year, (d-f) month of year, and (g-i) day of week. (a), (d) and (g) are Pearson's correlation coefficient ($r$); (b), (e) and (h) are fraction of data pairs within a factor of two (FAC2); and (c), (f) and (i) are normalised mean bias (NMB). Dots show individual site statistics ($n$ = 17, 30 and 2 for RB, UB and UT sites respectively), which are summarised in the superimposed box-plot whose shading demarcates the interquartile range (IQR) and whose whiskers extend to the largest and smallest value within 1.58 × IQR from the box hinges.





**Figure 4:** Model-measurement statistics per site for $PM_{10}$ daily mean concentrations during 2001–2010, by site type, and by (a–c) year, (d–f) month of year, and (g–i) day of week. (a), (d) and (g) are Pearson's correlation coefficient ($r$); (b), (e) and (h) are fraction of data pairs within a factor of two (FAC2); and (c), (f) and (i) are normalised mean bias (NMB) statistics ($n = 4$, 20 and 5 for RB, UB and UT sites respectively), which are summarised in the superimposed box-plot whose shading demarcates the interquartile range (IQR) and whose whiskers extend to the largest and smallest value within $1.58 \times$ IQR from the box hinges. Dots show individual site statistics. Whiskers extend to the largest and smallest value within $1.58 \times$ IQR from the box hinges.







**Figure 5:** Model-measurement statistics per site for PM$_{2.5}$ daily mean concentrations during 2009-2010, by site type, and by (a-c) year, (d-f) month of year, and (g-i) day of week. (a), (d) and (g) are Pearson's correlation coefficient ($r$); (b), (e) and (h) are fraction of data pairs within a factor of two (FAC2); and (c), (f) and (i) are normalised mean bias (NMB). Dots show individual site statistics ($n = 2$, 28 and 5 for RB, UB and UT sites respectively), which are summarised in the superimposed box-plot whose shading demarcates the interquartile range (IQR) and whose whiskers extend to the largest and smallest value within $1.58 \times$ IQR from the box hinges.