# Peer review of "Spatiotemporal evaluation of EMEP4UK-WRF v4.3 atmospheric chemistry transport simulations of health-related metrics for NO2, O3, PM10 and PM2.5 for 2001-2010"

_Geoscientific Model Development, 2016_

## Referee Comment (RC1) · Anonymous Referee #1 · 21 Sep 2016

The article presents a thorough evaluation of EMEP4UK model results against measurements of the AURN monitoring stations. While a thorough validation is a major and essential task when using an air quality model this article does not present any new insights or methodology on how such a validation should be done. Furthermore some of the presented validation work is IMO not complete and flawed to some extent. More specifically I have following remarks:

1) For some unclear reason the authors have omitted the root mean square (RMSE) statistic from their analysis. they base this e.a. on the Thunis et al., 2012 paper.

However in this paper even in the abstract the first statistic encountered is RMSE. In general there is agreement that a combination of bias, R and RMSE are best suited as each of these focuses on a different type of possible error in the model results when compared to observations.

2) On p. 13 line 10 - 14 the authors blame deviations between modeled and observed data (almost) completely on the observed data's lack on representativenes and measurement error. Problems in representativenes are rather a problem of incompatibility: both model and observations are representative at a certain scale (neither of which is better than the other). However, these scales could (and are often) incompatible but this is neither the fault of model or observation. Measurement error is indeed a concern but in practice model error often by far exceeds the measurement error.

2) In line with the previous remark, after reading the text I have some doubts on whether the authors have understood the full extent of the methodology presented to the FAIR-MODE community and outlined in the articles by Thunis et al. (2012) to which they refer. A sentence like p13 line 21" The presence of measurement certainty degrades the values that can be expected from air quality model measurement statistics" is a case in point: in the methodology proposed by Thunis et al. measurement uncertainty is used as the 'ruler' by which model uncertainty is judged: more measurement uncertainty then effectively means that model results can also be more uncertain!

In the end I was therefore left somewhat disconcerted by the text. Amassing all these results in a, admittedly, clear form must have been a major undertaking but there is not really anything new here. Worse yet, the authors seem to have missed some of the points made in the articles that they refer. I therefore recommend not publishing this article.
* * *

---

## Author Comment (AC1) · 30 Sep 2016

**"Spatiotemporal evaluation of EMEP4UK-WRF v4.3 atmospheric chemistry transport simulations of health-related metrics for $NO_2$, $O_3$, $PM_{10}$ and $PM_{2.5}$ for 2001–2010" by C. Lin et al.**

**Responses to anonymous reviewer #1**

We thank the reviewer for their time spent in reviewing our paper. Below, we respond to all comments made. The reviewer's comments are reproduced in their entirety, in italics.

*The article presents a thorough evaluation of EMEP4UK model results against measurements of the AURN monitoring stations. While a thorough validation is a major and essential task when using an air quality model this article does not present any new insights or methodology on how such a validation should be done. Furthermore some of the presented validation work is IMO not complete and flawed to some extent. More specifically I have following remarks:*

Response: We are pleased to read the reviewer's comment that our paper "presents a thorough evaluation of the EMEP4UK model results against measurements of the AURN monitoring stations." The reviewer then states that the article does not present any new insights or methodology on how such a validation should be done. In response we refer the reviewer and other readers of this discussion to the stated scope of Geoscientific Model Development, which encompasses articles reporting "full evaluations of previously published models" (http://www.geoscientific-model-development.net/about/aims_and_scope.html). Our article fits this scope: it reports, for the first time and for a temporally and spatially large dataset, the comparisons between output from the EMEP4UK model and observational data.

The criticisms in the latter part of the reviewer's comment above are repeated with more detail in their subsequent comments and we respond to them individually below.

*1) For some unclear reason the authors have omitted the root mean square (RMSE) statistic from their analysis. They base this e.a. on the Thunis et al., 2012 paper. However in this paper even in the abstract the first statistic encountered is RMSE. In general there is agreement that a combination of bias, R and RMSE are best suited as each of these focuses on a different type of possible error in the model results when compared to observations.*

Response: We are not clear why the reviewer thinks that our choice of the model-measurement statistics to present is based on the Thunis et al. (2012) paper. In our Introduction we cite several examples of the many studies that have discussed the choice of model-measurement statistic (for air quality studies), the work of Thunis and co-workers being amongst those we quote (P2, L5-9). We wrote: "Much has been written on air quality model evaluation (see, for example, Vautard et al., 2007; Dennis et al., 2010; Derwent et al., 2010; Rao et al., 2011; Thunis et al., 2012; Thunis et al., 2013; Pernigotti et al., 2013), including publications arising out of international collaborative programmes such as AQMEII (Air quality modelling evaluation international initiative, http://aqmeii-eu.wikidot.com) and FAIRMODE (Forum for air quality modelling in Europe, http://fairmode.jrc.ec.europa.eu)." These and other studies highlight the very wide suite of possible model-measurement statistics that can be used. We emphasise many times throughout our paper the basis of our selection of model-measurement comparison to publish in this paper (both the modelmeasurement statistics and the air pollutant concentration averaging used in those statistics): namely that it was guided by the needs of the health burden and epidemiology community making first use of this large model dataset. The first two sentences and the fourth sentence of our Abstract make this clear: "This study was motivated by the use in air pollution epidemiology and health burden assessment of data simulated at 5 km × 5 km horizontal resolution by the EMEP4UK-WRF v4.3 atmospheric chemistry transport model. Thus the focus of the model-measurement comparison statistics presented here was on the health-relevant metrics of annual and daily means of $NO_2$, $O_3$, $PM_{2.5}$ and $PM_{10}$ (daily maximum 8-hour running mean for $O_3$). "…"The two most important statistics highlighted in the literature for evaluation of air quality model output against policy (and hence health)-relevant standards – correlation and bias – were evaluated by site type, year, month and day-of-week." We do not dispute that RMSE is also a relevant model-measurement comparison statistic. But it is not practical to include results for all possible comparison statistics, which is why we focused on the correlation and bias statistics that are important for the health specialists. To further emphasise and justify this application of our evaluation we provided four paragraphs of discussion on the correlation and bias statistics in relation to health studies from P15, L1 to P15, L36. We will provide further emphasis and justification for our metrics in revised Introduction and Methods sections.

We refer to the work of Thunis et al. (2012) again in our Discussion section, in the context of commenting on the magnitudes of the model-measurement comparison statistics that may be expected for the type of air pollution model used in our work (see further comment on this below).

*2) On p. 13 line 10 - 14 the authors blame deviations between modeled and observed data (almost) completely on the observed data's lack on representativenes and measurement error. Problems in representativenes are rather a problem of incompatibility: both model and observations are representative at a certain scale (neither of which is better than the other). However, these scales could (and are often) incompatible but this is neither the fault of model or observation. Measurement error is indeed a concern but in practice model error often by far exceeds the measurement error.*

We believe the reviewer puts an interpretation on our text here that is not what we state, and at the same time ignores one of the key messages we promote from our model-measurement comparison. The specific text to which the reviewer refers above reads: "Even for a well-specified Eulerian model (in terms of input data, transport, chemistry, etc.), model-measurement agreement may not be perfect for (at least) the following two reasons: first, the model simulates a volume-averaged concentration whereas the monitor records the composition of the air in one part of that volume, which may or may not reflect the average concentration for the whole volume over the relevant time-averaging period; and, secondly, the measurement may be in error." So we and the reviewer are in agreement that there is an intrinsic incompatibility in the spatial scale of model and measurement. At no point here, or elsewhere in the paper, do we claim that one is better than the other, or 'blame' deviations between modelled and observed data "(almost) completely on measurements." We are simply reminding readers of this intrinsic incompatibility in scales, together with the reminder that measurements have an associated uncertainty. In fact, we do fully acknowledge model error at several points in our presentation and discussion of results, including in both the conclusions and in the abstract. We specifically emphasise (i.e. 'blame') shortcomings in emissions input into the model as being the dominant driver for the model-measurement

deviations (shortcomings in absolute magnitudes in emissions, in their temporal disaggregation and in the averaging of emissions across a model grid). For example, this is the text we write in the Abstract: "The directions of these biases are consistent with expectations of the effects of averaging primary emissions across the 5 km × 5 km model grid in urban areas, compared with monitor locations that are more influenced by these emissions than the grid average. …The biases are also indicative of potential underestimations of primary NOx and PM emissions in the model, and, for PM, with known omissions in the model of some PM components, e.g. wind-blown dust."; and, as further example, this is the text we write in the Conclusions "….is strongly indicative that the main driver of model shortcoming is inaccuracy of emissions (totals and the monthly and day-of-week temporal factors applied in the model to the totals)."

*2) In line with the previous remark, after reading the text I have some doubts on whether the authors have understood the full extent of the methodology presented to the FAIRMODE community and outlined in the articles by Thunis et al. (2012) to which they refer. A sentence like p13 line 21" The presence of measurement certainty degrades the values that can be expected from air quality model measurement statistics" is a case in point: in the methodology proposed by Thunis et al. measurement uncertainty is used as the 'ruler' by which model uncertainty is judged: more measurement uncertainty then effectively means that model results can also be more uncertain!*

Response: (We presume the reviewer intended to quote our text in their comment as "The presence of measurement UNcertainty degrades the values that can be expected from air quality model-measurement statistics", which is what we wrote, rather than "The presence of measurement certainty degrades the values…" which is what the reviewer writes that we wrote.) We don't understand why the reviewer thinks that we don't understand the concept that the greater the uncertainty that may exist in measurements the poorer the model-measurement comparison statistics may be. We think our sentence fully encapsulates this concept. We refer to the work of Thunis and co-workers at this point in the Discussion as a very useful previously-published 'yard stick' for the magnitudes of correlation coefficients and bias that might be expected for atmospheric chemistry transport model output vs. measurement (which are of similar construct to our model-measurement comparisons) when allowing for the possibility that there may be uncertainty in the measurement up to the level permitted under EU directives for reporting air pollutant measurements. We do not claim that these levels of uncertainties are the actual uncertainties in our particular set of measurements, but that if they were then these are the sorts of magnitudes of model-measurement statistics that might be expected.

*In the end I was therefore left somewhat disconcerted by the text. Amassing all these results in a, admittedly, clear form must have been a major undertaking but there is not really anything new here. Worse yet, the authors seem to have missed some of the points made in the articles that they refer. I therefore recommend not publishing this article.*

Response: We hope that our extensive responses above have addressed the reviewer's concerns. In summary, the novelty of work is the publication of new model evaluation statistics derived from an extensive set of simulations from the EMEP4UK model, with deliberate focus on the model-measurement comparison needs of the health burden and epidemiology community users of these simulations.

References cited in this response

Dennis, R., Fox, T., Fuentes, M., Gilliland, A., Hanna, S., Hogrefe, C., Irwin, J., Rao, S. T., Scheffe, R., Schere, K., Steyn, D. and Venkatram, A.: A framework for evaluating regional-scale numerical photochemical modeling systems, Environmental Fluid Mechanics, 10, 471-489, 2010.

Derwent, D., Fraser, A., Abbott, J., Jenkin, M. E., Willis, P. and Murrells, T.: Evaluating the performance of air quality models, A report for Defra and the Devolved Administrations, http://www.airquality.co.uk/reports/cat05/1006241607_100608_MIP_Final_Version.pdf, 2010.

Pernigotti, D., Gerboles, M., Belis, C. A. and Thunis, P.: Model quality objectives based on measurement uncertainty. Part II: NO2 and PM10, Atmos. Environ., 79, 869-878, 2013.

Rao, S. T., Galmarini, S. and Puckett, K.: Air Quality Model Evaluation International Initiative (AQMEII) Advancing the State of the Science in Regional Photochemical Modeling and Its Applications, Bulletin of the American Meteorological Society, 92, 23-30, 2011.

Thunis, P., Pederzoli, A. and Pernigotti, D.: Performance criteria to evaluate air quality modeling applications, Atmos. Environ., 59, 476-482, 2012.

Thunis, P., Pernigotti, D. and Gerboles, M.: Model quality objectives based on measurement uncertainty. Part I: Ozone, Atmos. Environ., 79, 861-868, 2013.

Vautard, R., Builtjes, P. H. J., Thunis, P., Cuvelier, C., Bedogni, M., Bessagnet, B., Honore, C., Moussiopoulos, N., Pirovano, G., Schaap, M., Stern, R., Tarrason, L. and Wind, P.: Evaluation and intercomparison of Ozone and PM10 simulations by several chemistry transport models over four European cities within the CityDelta project, Atmos. Environ., 41, 173-188, 2007.

---

## Referee Comment (RC2) · Anonymous Referee #1 · 11 Oct 2016

1) There are indeed many statistics some of which are 'quite exotic' that can be used to assess model performance so one could argue that RMSE should not be included. However, in all articles quoted RMSE is included and this is not just because the authors had a big appetite for statistics but because this statistic 'completes the picture' when assessing model performance in conjunction with the bias and correlation. Why not extend your analysis with the RMSE?

2) There was indeed a typo in my comment: this should off course have been 'UNcertainty'. However the response the authors provide to the comment I gave concerning

model vs observation uncertainty rather me in my conviction that they did not fully understand the concept proposed by Thunis et al. Let me try to explain. The concept of a Model Quality Objective (MQO) presented by Thunis et al. is that statistics used to describe model performance (bias, R or RMSE, ...) in themselves do not allow an actual assessment of how good the model is performing. Thunis et al. therefore propose to use the observation uncertainty as a 'yard stick' by which model uncertainty can be assessed. This e.a. implies that if model uncertainty is smaller than observation uncertainty there is no statistical basis for trying to improve the model in the sense that you'll not be able to discern the improvement based on a comparison with measured values. This also means that if measurement uncertainty increases this does not 'degrade the values' but rather result in that 'poorer' model performance may still be acceptable.

---

## Referee Comment (RC3) · Anonymous Referee #2 · 22 Nov 2016

The paper presents a thorough and well laid out evaluation of the performance of the EMEP4UK-WRF model by comparison with observations from the AURN network based on metrics most appropriate to assessment of health impacts. The assessment is thorough and results are presented for a range of station types, for different averaging periods and for a range of pollutants. Explanations are given and discussed for discrepancies between modelled and observed values, such as the model overestimation of O3 and underestimation of NO2. I believe the paper fits the remit of the journal, as set out in the GMD Aims and Scope, and I recommend some minor revisions be made as follows:

[Figure]

Introduction, page 2, line 27-29. The text is slightly confusing since the authors suggest they have undertaken epidemiological studies, although the current paper is not based on epidemiology, rather it is atmospheric chemistry modelling. If the authors are referring to work other than this paper, references should be given at the end of the sentence, or the text made clearer as to what is being referred to here.

Page 3, line 17: please clarify daily mean as 24 hour mean here to remove any potential ambiguity. Section 2.4, page 6: The FAC2 metric is explained here but has been presented earlier in the paper without enough explanation (in Table 1 for example). Introduction, line 4: Suggest add references to COMEAP 2009 report for PM. Line 18: suggest replace "and away from" with "or away from". Page 14, line 17: "trends" not "trend"

Table 1: Some of the abbreviations need expanding (e.g. FAC2) in the table heading since they are not addressed previously in the text.

———————————————

---

## Author Comment (AC2) · 15 Dec 2016

gmd-2016-183
**"Spatiotemporal evaluation of EMEP4UK-WRF v4.3 atmospheric chemistry transport simulations of health-related metrics for $NO_2$, $O_3$, $PM_{10}$ and $PM_{2.5}$ for 2001–2010" by C. Lin et al.**

**Responses to anonymous reviewer #1's second comments (RC2)**

We thank the reviewer for their additional comments, to which we respond below. The reviewer's comments are reproduced in italics.

*1) There are indeed many statistics some of which are 'quite exotic' that can be used to assess model performance so one could argue that RMSE should not be included. However, in all articles quoted RMSE is included and this is not just because the authors had a big appetite for statistics but because this statistic 'completes the picture' when assessing model performance in conjunction with the bias and correlation. Why not extend your analysis with the RMSE?*

Response: We do not dispute that RMSE is also an often-used model-measurement statistic, but we focused here on the correlation and bias statistics relevant for the health effects community using the output from these model simulations – as is stated in the abstract, with detailed epidemiological commentary on this in the Discussion section. (See also our response to the similar comment in our upload to the online Interactive Discussion section of this paper on 30/09/16.) Reviewer #2 accepts the appropriateness of our material. Our paper contains a large number of tables and graphics of model-measurement statistics already. All the raw model and measurement values used to calculate our statistics will be available to allow anyone to calculate any additional model-measurement statistics.

In the revised paper we have provided additional up-front confirmation of the model-measurement statistics computed in this work with the addition of the following sentence at the end of the Introduction: "Two important statistics for evaluation of air quality model output for health studies – correlation and bias (see Discussion) – were evaluated by type of monitor location, year, month and day-of-week." We have also changed the phrasing from "the two most important statistics" to "two important statistics" where similar text occurs elsewhere in the paper.

*2) There was indeed a typo in my comment: this should off course have been 'UNcertainty'. However the response the authors provide to the comment I gave concerning model vs observation uncertainty rather me in my conviction that they did not fully understand the concept proposed by Thunis et al. Let me try to explain. The concept of a Model Quality Objective (MQO) presented by Thunis et al. is that statistics used to describe model performance (bias, R or RMSE, ...) in themselves do not allow an actual assessment of how good the model is performing. Thunis et al. therefore propose to use the observation uncertainty as a 'yard stick' by which model uncertainty can be assessed. This e.a. implies that if model uncertainty is smaller than observation uncertainty there is no statistical basis for trying to improve the model in the sense that you'll not be able to discern the improvement based on a comparison with measured values. This also means that if measurement uncertainty increases this does not 'degrade the values' but rather result in that 'poorer' model performance may still be acceptable.*

Response: We thank the reviewer for providing further explanation of the concept of the model quality objective presented by Thunis and co-workers. We maintain that we understand the

concept, and that the issue is in the wording in some of the text we used to expresses our thoughts. We understand that the measurement uncertainty is used as a 'yard stick' against which model uncertainty can be assessed and that if model uncertainty is smaller than measurement uncertainty then it will not be possible to discern any improvement in model performance, when model performance is being assessed against measurements. We accept that the use of the phrasing "degrade the values" in our original statement – "The presence of measurement uncertainty degrades the values that can be expected from air quality model-measurement statistics" – is misleading. We have now amended this sentence to read: "The presence of measurement uncertainty constrains the extent to which model-measurement statistics can be used to evaluate the performance of a model." We have also amended the first sentence in the following paragraph from the original: "Table 3 shows that in the large majority of instances the values of model-measurement correlation and NMB from this EMEP4UK-WRF modelling exceed the threshold values described above for satisfactory model performance in the presence of measurement uncertainties at the levels assigned." to now read: "Table 3 shows that in the large majority of instances the values of model-measurement correlation and NMB from this EMEP4UK-WRF modelling satisfy the model performance criteria values derived for measurement uncertainties at the magnitudes discussed above." Phrasing in the relevant sentence in the abstract has also been modified to now read: "Model-measurement correlation and bias were generally better than values that incorporate realistic magnitudes of measurement uncertainties." We have thoroughly re-read all text and believe there is nothing that is incompatible with the work of Thunis and co-workers.

---

## Author Comment (AC3) · 15 Dec 2016

gmd-2016-183
**"Spatiotemporal evaluation of EMEP4UK-WRF v4.3 atmospheric chemistry transport simulations of health-related metrics for $NO_2$, $O_3$, $PM_{10}$ and $PM_{2.5}$ for 2001–2010" by C. Lin et al.**

**Responses to anonymous reviewer #2 (RC3)**

We thank the reviewer for their time spent reviewing our paper. Our responses to the comments made are given below. The reviewer's comments are reproduced in italics here.

*The paper presents a thorough and well laid out evaluation of the performance of the EMEP4UK-WRF model by comparison with observations from the AURN network based on metrics most appropriate to assessment of health impacts. The assessment is thorough and results are presented for a range of station types, for different averaging periods and for a range of pollutants. Explanations are given and discussed for discrepancies between modelled and observed values, such as the model overestimation of O3 and underestimation of NO2. I believe the paper fits the remit of the journal, as set out in the GMD Aims and Scope, and I recommend some minor revisions be made as follows:*

Response: We thank the reviewer for their endorsement of the thoroughness and appropriateness of the material we present in the paper, and for their recommendation of its suitability for publication in GMD. We have made all requested minor revisions as indicated below.

*Introduction, page 2, line 27-29. The text is slightly confusing since the authors suggest they have undertaken epidemiological studies, although the current paper is not based on epidemiology, rather it is atmospheric chemistry modelling. If the authors are referring to work other than this paper, references should be given at the end of the sentence, or the text made clearer as to what is being referred to here.*

Response: We have both rephrased this sentence to make its message more direct and added a citation to an epidemiological study using these modelling data. The modified sentence now reads (page 2, lines 27-29): "As part of a multi-institution project on the health impacts of exposure to multiple pollutants, we have derived UK-wide distributions of surface air pollution at hourly temporal resolution over multiple years (2001-2010), at 5 km × 5 km horizontal resolution, using the EMEP4UK-WRF atmospheric chemistry transport model (ACTM) (Butland et al., 2016)."

*Page 3, line 17: please clarify daily mean as 24 hour mean here to remove any potential ambiguity.*

Response: The text has been modified at this point to "daily (i.e. 24-h) mean" so as to emphasise to the reader that the use of the phrase "daily mean" throughout this paper refers to the full 24-h mean.

*Section 2.4, page 6: The FAC2 metric is explained here but has been presented earlier in the paper without enough explanation (in Table 1 for example).*

Response: We have checked and Table 1 is the only place where the terminology 'FAC2' appears before its definition is given in Section 2.4. Also, although 'FAC2' is one of the

column headings in Table 1, discussion of the data in this column of the table does not occur until the Results section of the paper, after the definition of FAC2 has been provided. Nevertheless, we appreciate the reviewer's comment that the reader's attention is drawn to Table 1 earlier in the paper and that this could cause confusion. We have therefore inserted the additional text "(as defined in Section 2.4)" after the text in the caption of Table 1 that states that the table also contains a summary of measurement-model statistics.

*Introduction, line 4: Suggest add references to COMEAP 2009 report for PM.*

Response: The COMEAP (2009) reference has been added to the citations in this sentence.

*Line 18: suggest replace "and away from" with "or away from".*

Response: The requested change has been made.

*Page 14, line 17: "trends" not "trend"*

Response: The requested change has been made.

*Table 1: Some of the abbreviations need expanding (e.g. FAC2) in the table heading since they are not addressed previously in the text.*

Response: The definition for the acronym AURN, the UK Automatic Urban and Rural Network, is now given in full in the caption to Table 1. For the meanings of the acronyms for the measurement-model statistics presented in Table 1 the reader is now directed via the table caption to Section 2.4 where the definitions of these statistics is given (see also response to a comment above).

References cited in this response

Butland, B. K., Atkinson, R. W., Milojevic, A., Heal, M. R., Doherty, R. M., Armstrong, B. G., MacKenzie, I. A., Vieno, M., Lin, C. and Wilkinson, P.: Myocardial infarction, ST-elevation and non ST-elevation myocardial infarction and modelled daily pollution concentrations: a case-crossover analysis of MINAP data, Open Heart, 3, e000429. doi:10.1136/openhrt-2016-000429, 2016.

COMEAP: Long-term exposure to air pollution: effect on mortality, UK Department of Health Committee on the Medical Effects of Air Pollutants. ISBN 978-0-85951-640-2, https://www.gov.uk/government/publications/comeap-long-term-exposure-to-air-pollution-effect-on-mortality, 2009.

---

## Referee Report (RR1)

Review of (gmd-2016-183): Spatiotemporal evaluation of EMEP4UK-WRF atmospheric chemistry transport simulations of health-related metrics for $NO_2$, $O_3$, $PM_{10}$ and $PM_{2.5}$ for 2001-2010 by C. Lin et al.

The Authors present in this work a thorough evaluation of the EMEP4UK-WRF model system applied over the UK for a series of meteorological years. This is a challenging task and the Authors succeeded to present it in a systematic and organised manner (different time periods, pollutants…).  Having read the two other Reviewer's comments and the response of the Authors to those comments, I would make the following two comments.

1)  I agree with Reviewer one regarding the use of the RMSE. Even though the Authors state that the correlation and bias are the two most appropriate statistics given their health oriented purpose, I believe it is important to add RMSE to those two statistics. It is particularly important in the discussion section where the values reported by Thunis et al. (2012) are used to judge the quality of the EMEP4UK-WRF results. It is clearly stated in Thunis et al. that the fulfilment of the criteria on bias, correlation and standard deviation is a necessary but not sufficient condition to assess the quality of the model results, and that the RMSE remains the key indicator to do this. I would therefore encourage the Authors to add this statistics to their work. I would also suggest them to use the latest uncertainty parameter values as reported in the Fairmode documents (available on the web portal).

2)  The use of the RMSE indicator would certainly clearly show that the traffic stations should not be used in this evaluation. Many published works have shown the inadequacy of a 5x5 km resolution model to capture street concentrations, especially for $O_3$ or $NO_2$. I believe these stations should be withdrawn at start from this work. The Authors refer to the underestimation of local scale emissions but these issues are well known and keeping these traffic stations together with the others is confusing for this type of model application.

In conclusion I believe this work is worth publishing but some major revisions would be needed.

---

## Author Response (AR2)

gmd-2016-183
**"Spatiotemporal evaluation of EMEP4UK-WRF v4.3 atmospheric chemistry transport simulations of health-related metrics for $NO_2$, $O_3$, $PM_{10}$ and $PM_{2.5}$ for 2001–2010" by C. Lin et al.**

**Responses to referee report 1**

We thank the additional referee for their time spent reviewing the paper and our previous responses to review comments. We welcome the referee's final comment that the work is worth publishing after undertaking the further revisions suggested.

Our responses to this referee's comments are given below with their comments reproduced in italics.

*The Authors present in this work a thorough evaluation of the EMEP4UK-WRF model system applied over the UK for a series of meteorological years. This is a challenging task and the Authors succeeded to present it in a systematic and organised manner (different time periods, pollutants…).*

Response: We are grateful for these supportive comments on our presentation of the large dataset of model-measurement comparisons.

*Having read the two other Reviewer's comments and the response of the Authors to those comments, I would make the following two comments.*
*1) I agree with Reviewer one regarding the use of the RMSE. Even though the Authors state that the correlation and bias are the two most appropriate statistics given their health oriented purpose, I believe it is important to add RMSE to those two statistics. It is particularly important in the discussion section where the values reported by Thunis et al. (2012) are used to judge the quality of the EMEP4UK-WRF results. It is clearly stated in Thunis et al. that the fulfilment of the criteria on bias, correlation and standard deviation is a necessary but not sufficient condition to assess the quality of the model results, and that the RMSE remains the key indicator to do this. I would therefore encourage the Authors to add this statistics to their work. I would also suggest them to use the latest uncertainty parameter values as reported in the Fairmode documents (available on the web portal).*

Response: In our revised paper we now include the model-measurement RMSE statistics alongside the model-measurement correlation and bias statistics. Thus all of Figures 2-5 now contain additional panels illustrating the distributions of individual-site RMSE statistics. Tables 1 and 3 likewise now include an additional column that summarises the RMSE data alongside the similar summaries of the correlation and bias data. The text has been edited throughout to state that model-measurement RMSE statistics are included and to highlight relevant observations from the RMSE statistics alongside observations on the other model-measurement statistics.

We have also revised again and extended the text in our paper discussing the comparison of our model-measurement statistics with the values of model performance criteria (MPC) developed from the FAIRMODE project from consideration of uncertainty in air pollutant measurements and the AIRBASE database of measured European air pollutant concentrations (page 13, line 21 to page 14, line 22). As per a previous response to a review comment we accept that some of our initial phrasing was misleading. We believe we do not now write anything that is incompatible with the above. In particular we now include the explicit statement that "satisfying the MPC is a necessary but not sufficient part of model validation" (page 13, line 39). We have also examined again the further published papers on this topic from the FAIRMODE project (Thunis et al., 2013; Pernigotti et al., 2013), and documents on the FAIRMODE website, which collectively include more detailed and updated evaluation of potential measurement uncertainty as a function of the concentration being measured, rather than assuming a constant relative uncertainty at the level specified in the EU Directive at the limit value. We have now included the updated values for MPC for $O_3$ and $PM_{10}$ in our Table 3 and in the discussion. We include the statement that "The

intention here is to provide an overview of how the EMEP4UK-WRF model-measurement statistics compare in general with the threshold criteria for comparison of an air quality model against measurement in the European air quality context", i.e. we are not undertaking a forensic examination. We also note that the FAIRMODE project has undertaken detailed evaluation of potential levels of measurement uncertainty for $PM_{2.5}$ and $NO_2$ (for hourly average, rather than daily average, in the case of $NO_2$) but that estimated values of MPC for daily mean $PM_{2.5}$ and daily mean $NO_2$ using a comparable measurement dataset to those for $O_3$ and $PM_{10}$ are not published.

*2) The use of the RMSE indicator would certainly clearly show that the traffic stations should not be used in this evaluation. Many published works have shown the inadequacy of a 5x5 km resolution model to capture street concentrations, especially for O3 or NO2. I believe these stations should be withdrawn at start from this work. The Authors refer to the underestimation of local scale emissions but these issues are well known and keeping these traffic stations together with the others is confusing for this type of model application.*

Response: As requested by the reviewer we have now entirely removed all model-measurement comparison data associated with traffic stations. Thus all of Figures 1-5 and Tables 1-3 have been modified to reflect the removal of these evaluations and the text has been edited accordingly throughout. Similar modifications have been made to the additional figures and tables in the Supplementary Information, which is now uploaded afresh.

*In conclusion I believe this work is worth publishing but some major revisions would be needed.*

Response: We thank for the referee for this support. We have undertaken all the revisions requested.

References cited

[revised manuscript text omitted]

(a)                    (b)          (c)          (d)

(e)                    (f)          (g)          (h)

(i)           (j)          (k)                    (l)

---

## Author Response (AR3)

gmd-2016-183
**"Spatiotemporal evaluation of EMEP4UK-WRF v4.3 atmospheric chemistry transport simulations of health-related metrics for NO₂, O₃, PM₁₀ and PM₂.₅ for 2001–2010" by C. Lin et al.**

Note that the track-changed version of the manuscript provided below shows all changes with respect to the original first submission of this paper published as the Discussion article in 2016 (not just the further changes with respect to the first revision submitted in February 2017).

**Responses to Anonymous Referee #3 second report**

The referee's comments are given below in italics.

*The document has improved significantly and all suggested revisions have been accounted for.*
*I would only propose a couple of minor comments to complement the analysis:*

Response: We thank the reviewer for confirming that the revised document we previously uploaded satisfied all previous reviewer requests. The reviewer has made suggestions for two further analyses and we confirm here that we have undertaken both of these – details below.

*1) In figure one, the fac2 lines could be complemented by additional lines indicating the yearly averaged model quality objectives (MQO) as a function of the concentration. An example of application can be found in the report (see example p34)*
*http://fairmode.jrc.ec.europa.eu/document/fairmode/WG1/Guidance_MQO_Bench_vs2.1.pdf*
*To fulfill the MQO, 90% of the stations points should lay between the MQO lines.*

Response: The reviewer has made a useful further suggestion. We have applied the formulae and variable values given in the above-cited report to the measurement data used in our model-measurement comparisons to calculate the relevant MQO values. We have added these MQO values as an additional column to Table 2 and, as suggested by the reviewer, plotted them as additional lines (in green) on updated versions of each of the 8 panels in Figure 1. We have also provided the number and % of sites in each panel that satisfy this model performance indicator as another column in Table 2. A new paragraph of text describing these new data and their interpretation has been added to the Discussion (beginning L15 on P14).

*2) In the same report, uncertainty values can be found for PM2.5. These values could be added to the values in table 3 to complement the values already available for O3 and PM10*

Response: The reviewer suggests that we use the formulae and uncertainty values in the Fairmode report to calculate values of minimum performance criteria for $r$, NMB and RMSE metrics for daily mean PM₂.₅. We have done this using our dataset of measurement values. We have now also undertaken the calculations of values of minimum performance criteria for $r$, NMB and RMSE for daily mean PM₁₀ and daily maximum 8-h mean O₃ using our dataset of measurement values. Previously we had not undertaken our own calculations of model performance criteria for PM₁₀ and O₃ but had quoted in Table 3 example values from the literature for these model performance criteria. This means that the model performance values given in the right hand columns of Table 3 are now all updated to be based on calculations on measurement values relevant to this study, as well as to include the new values for PM₂.₅. In the Table we provide model performance criteria values calculated both with the assumption of constant relative uncertainty in each measurement, and with the assumption of a concentration-dependent uncertainty in each measurement. As well as the updates to Table 3, the Discussion text (and captions and footnotes to Table 3) have all been amended to describe the new set of model performance criteria values. It is important to stress, however, that whilst the absolute numbers in Table 3 have changed somewhat, the overall observation has not; namely that 
[revised manuscript text omitted]